# Erythropoietin *N*-glycosylation of Therapeutic Formulations Quantified and Characterized: An Interlab Comparability Study of High-Throughput Methods

**DOI:** 10.3390/biom14010125

**Published:** 2024-01-18

**Authors:** Róisín O’Flaherty, Manuela Amez Martín, Richard A. Gardner, Patrick M. Jennings, Pauline M. Rudd, Daniel I. R. Spencer, David Falck

**Affiliations:** 1National Institute for Bioprocessing, Research and Training, Fosters Avenue, Blackrock, A94 X099 Dublin, Irelandpjenning@amgen.com (P.M.J.);; 2Department of Chemistry, Maynooth University, W23 F2K8 Maynooth, Ireland; 3Ludger Ltd., Culham Science Centre, Abingdon OX14 3EB, UK; m.amez_martin@lumc.nl (M.A.M.); richard.gardner@ludger.com (R.A.G.); daniel.spencer@ludger.com (D.I.R.S.); 4Center for Proteomics and Metabolomics, Leiden University Medical Center, Albinusdreef 2, 2333 ZA Leiden, The Netherlands

**Keywords:** erythropoietin, EPO glycosylation, *N*-glycans, hydrophilic interaction chromatography, mass spectrometry, interlaboratory study

## Abstract

Recombinant human erythropoietin (EPO) is a biopharmaceutical frequently used in the treatment of anemia. It is a heavily glycosylated protein with a diverse and complex glycome. EPO *N*-glycosylation influences important pharmacological parameters, prominently serum half-life. Therefore, EPO *N*-glycosylation analysis is of the utmost importance in terms of controlling critical quality attributes. In this work, we performed an interlaboratory study of glycoanalytical techniques for profiling and in-depth characterization, namely (1) hydrophilic interaction liquid chromatography with fluorescence detection after 2-aminobenzamide labeling (HILIC-FLD(2AB)) and optional weak anion exchange chromatography (WAX) fractionation and exoglycosidase digestion, (2) HILIC-FLD after procainamide labeling (PROC) optionally coupled to electrospray ionization-MS and (3) matrix-assisted laser desorption ionization time-of-flight mass spectrometry (MALDI-MS). All techniques showed good precision and were able to differentiate the unique *N*-glycosylation profiles of the various EPO preparations. HILIC-FLD showed higher precision, while MALDI-TOF-MS covered the most analytes. However, HILIC-FLD differentiated isomeric *N*-glycans, i.e., *N*-acetyllactosamine repeats and *O*-acetylation regioisomers. For routine profiling, HILIC-FLD methods are more accessible and cover isomerism in major structures, while MALDI-MS covers more minor analytes with an attractively high throughput. For in-depth characterization, MALDI-MS and HILIC-FLD(2AB)/WAX give a similar amount of orthogonal information. HILIC-FLD(PROC)-MS is attractive for covering isomerism of major structures with a significantly less extensive workflow compared to HILIC-FLD(2AB)/WAX.

## 1. Introduction

Recombinant human erythropoietin (EPO) has important therapeutic applications, for example in treating anemia in chronic kidney disease and cancer [1,2]. EPO contains *N-* and *O*-glycosylation. *N*-glycosylation makes up roughly half of EPO’s molecular weight. It is critical for the quality, safety and potency of EPO [3]. Consequently, several glycosylation traits are considered to be critical quality attributes (CQAs) [4]. EPO is also one of the most well-known examples of successful glycoengineering [5]. In conclusion, the analysis of EPO *N*-glycosylation is of the utmost importance to the biopharmaceutical industry.

Epoetin alpha, initially Eprex and now many biosimilars, was the first generation of therapeutic EPO [3,6]. The second generation, epoetin beta and epoetin delta, sought to improve EPO quality through an altered glycosylation profile. Finally, the third generation, darbepoetin alfa, increased the number of *N*-glycosylation sites from three to five by only five mutations to the protein backbone, which drastically improved EPO half-life [7]. In this study, all these three generations are represented, Eprex being an epoetin alpha, a European Pharmacopoeia standard (PharmEPO) being a mixture of epoetin alpha and epoetin beta and Aranesp being a darbepoetin alfa.

Due to its importance, the analysis of EPO *N*-glycosylation has been reviewed extensively [3,8,9]. In addition to analysis at the intact protein [10] and glycopeptide level [11,12], EPO *N*-glycosylation is frequently analyzed on the released glycan level. Intact protein and glycopeptide level analysis additionally allow assessment of the single *O*-glycosylation site of EPO [3]. EPO *N*-glycosylation analysis on the released glycan level will be the focus of this method comparison. Most frequently, the enzyme PNGase F is employed to obtain the fully intact glycan as a glycosylamine species or, after hydrolysis, with a free reducing end. This can be analyzed as such or after labeling with a fluorophore, through carbamate chemistries or reductive amination most commonly [13,14]. Four methods are dominantly used to analyze released *N*-glycans. Firstly, the most established, gold standard method is fluorescence detection (FLD) after separation with hydrophilic interaction liquid chromatography (HILIC) [15]. FLD is generally considered to be the most robust and repeatable detection for quantitation. Secondly, HILIC separation can be combined with mass spectrometry (MS), providing additional selectivity [16]. Thirdly, MS may have enough selectivity on its own, so separation may be omitted. For this approach, matrix-assisted desorption/ionization-time-of-flight-MS (MALDI-MS) is an attractive approach [17]. Permethylation has long been the derivatization method of choice for MALDI-MS analysis [18], but the challenges of controlling the specificity of the highly reactive reagents have sparked interest in alternative strategies for the stabilization and neutralization of sialic acids [17]. Approaches based on separation through the use of capillary electrophoresis (CE), more specifically by capillary zone electrophoresis or capillary gel electrophoresis, which may be combined with FLD or MS, demonstrate suitable alternative approaches but are not assessed in this particular interlab study [19,20].

Some of the advantages and disadvantages of these techniques are well known. However, the extent to which many of these attributes manifest and aid or impair the analysis greatly depends on the actual composition of the glycome in question. EPO’s glycome is especially challenging due to the large diversity of highly variable features, such as antennarity (2 to 4), sialylation (1 to 4), LacNAc-repeats (0 to 5), *N*-glycolylneuraminic acid (NeuGc; typically a few percent of the sialic acid (SA) content), sialic acid *O*-acetylation (0 to 2 per SA), etc. [3]. A comparison of these methods, even specifically for EPO, may have been possible through a literature study. However, it is our experience that a comparison based on specifically acquired data has much added value and is greatly appreciated by the biopharmaceutics community [21]. The experimenter and the laboratory can be strong confounders of an analysis which presents a significant hurdle for any comparison of data. Partially conversely, random and systematic errors are strongly influenced by experience and quality standards. At least in glycomics, experienced laboratories/experimenters working to high standards achieve surprisingly comparable results [21,22,23]. A single experimenter/laboratory cannot gather sufficient expertise in all techniques to perform all to the highest standards. Hence, our preferred approach to compare methods, as used herein, is for each technique to be performed by the most qualified laboratory. Each partner had extensive experience in the respective technique, including the analysis of EPO *N*-glycosylation.

We compared these frequently used methods focusing on three challenges, which would typically be encountered in a high-throughput (HT) biopharmaceutical analysis: (1) to produce a unique fingerprint of each product with high repeatability, (2) to deliver structural resolution so profile changes can be linked to functional aspects and (3) inter-method comparability of the results, specifically focusing on CQAs. The *N*-glycans released from the three biopharmaceutical EPO samples were analyzed through the use of three techniques, a total of five methods, namely (1) HILIC-FLD(2AB) with and without prior weak anion exchange liquid chromatography (WAX) fractionation and exoglycosidase digestion, (2) HILIC-FLD(PROC) optionally coupled to electrospray ionization-MS and (3) MALDI-MS. For profiling, HILIC-FLD(2AB), HILIC-FLD(PROC) and MALDI-MS were compared, while deep characterization was achieved through the use of HILIC-FLD(2AB)/WAX, HILIC-FLD(PROC)-MS and MALDI-MS.

## 2. Materials and Methods

### 2.1. Samples

The study consisted of the analysis of three different samples, each measured in quintuplicate through the use of different methods in independent laboratories. Two biopharmaceutical formulations were sourced through the hospital pharmacy. Eprex^®^ (Jansen Biologics, Leiden, The Netherlands) is an epoetin alfa product and Aranesp^®^ (Amgen Europe, Breda, The Netherlands) is a darbepoetin alfa product. A third sample, an erythropoietin standard for physicochemical tests (PharmEPO), was obtained from the European Pharmacopoeia of the European Directorate for the Quality of Medicines & Healthcare (Strasbourg, France). Dosages from a single batch could be obtained for Aranesp^®^ and PharmEPO. For Eprex^®^, dosages from two different batches were mixed, and aliquots of this master mix were sent to the participating laboratories.

### 2.2. Matrix Assisted Laser Desorption/Ionization-Time-of-Flight-Mass Spectrometry (MALDI-MS)

This analysis was performed as described earlier [17]. A preparation aliquot containing 5 µg of EPO was diluted in 100 mM formic acid (Merck KGaA, Darmstadt, Germany) to a volume of 100 µL incubated for 15 min at room temperature and freeze-dried. After reconstitution in 10 μL phosphate-buffered saline solution (PBS; Merck KGaA), which contained 1% NP-40 (Merck KGaA) and 0.5 U PNGaseF (Roche Diagnostics, Mannheim, Germany), samples were incubated for circa 18 h to release *N*-glycans. Afterward, precipitation of proteins and sialic acid derivatization (lactonization of α2,3-linked sialic acids) was performed by adding a 250 mM solution of 1-ethyl-3-(3-(dimethylaminopropyl)carbodiimide hydrochloride and 1-hydroxybenzotriazole hydrate in 100 μL ethanol (all Merck KGaA). *N*-glycans were purified via HILIC-solid phase extraction (SPE): pipette tips were filled with cotton thread, washed three times with 20 μL water, and equilibrated three times with 20 μL 85% acetonitrile (ACN, Biosolve, Valkenswaard, The Netherlands); samples were captured by 20 times aspiration, and washed three times each with 20 μL 85% ACN containing 1% trifluoroacetic acid and with 20 μL 85% ACN. After elution in 10 μL water, 4 μL of sample were mixed with 1 μL of superDHB (a 9:1 mixture of 2,5-dihydroxybenzoic acid and 2-hydroxy-5-methoxybenzoic acid at 5 mg/mL in 50% ACN, Merck KGaA) containing 1 mM sodium hydroxide and crystalized on an 800/384 AnchorChip MALDI target plate (Bruker Daltonics, Bremen, Germany). Analysis occurred in reflectron positive ion mode on an UltrafleXtreme MALDI-TOF-MS (Bruker Daltonics) measuring between *m*/*z* 1000 and *m*/*z* 5000. Spectra were automatically processed using MassyTools (0.1.8.1.2, ‘open source’, Leiden, The Netherlands ) as previously described [17,24].

### 2.3. Hydrophilic Interaction Liquid Chromatography-Fluorescence Detection after 2-Aminobenzamide Labeling (HILIC-FLD(2AB))

Sample preparation was performed on a Hamilton Robotics StarLet liquid handling platform (Hamilton Company, Bonaduz, Switzerland) similar to previous descriptions [25]. An equivalent of 20 µg of EPO was dried down, reconstituted in 5 µL water and reduced and denatured in 100 mM ammonium bicarbonate and 12 mM dithiothreitol at 65 °C for 30 min. After cooling for 10 min, the samples were alkylated with 17 mM iodoacetamide at room temperature for 30 min. Consecutive incubation at 37 °C for 120 min and at boiling for 10 min with 400 U of trypsin (Pierce, ThermoFisher Scientific, Waltham, MA, USA) achieved proteolytic cleavage. Glycans were released with 0.5 mU PNGase F (formerly ProZyme, Agilent, Santa Clara, CA, USA) at pH 8 for 120 min at 40 °C.

Hydrazide-assisted glycan clean-up and glycan labeling followed. In brief, released glycans were captured by rebuffering 40 µL of Ultralink hydrazide (ThermoFisher Scientific) resin suspension to acetonitrile (ACN) containing 2% acetic acid and adding 10% sample. After incubation at 70 °C for 70 min, the resin was washed with methanol (MeOH), 2M guanidine, water, MeOH with 1% triethylamine and MeOH in sequence. Further incubation with MeOH: acetic anhydride 90:10 for 30 min at room temperature preceded removal of excess reagent via filtration and washing with MeOH, water and ACN, respectively. Resin was reconstituted in 88% ACN with 2% acetic acid at 70 °C for 90 min. Glycans were fluorescently labeled at 65 °C for 120 min under these final conditions: 70% ACN, 8% water, 14% dimethyl sulfoxide (DMSO), 7.3% acetic acid, 200 mM sodium cyanoborohydride and 70 mM 2-aminobenzamide (2AB).

Final purification via SPE proceeded via the quenching of the reaction with 95% ACN. Resin supernatant was trapped on a HyperSep Diol SPE cartridge (ThermoFischer Scientific) for 5 min and washed three times with 95% acetonitrile. 2AB-labeled glycans were vacuum eluted with 20% ACN, dried, re-dissolved in 70% ACN and filtered.

HILIC-FLD(2AB) analysis of labeled glycans, also reported previously,[25] was achieved with an Ethylene Bridged Hybrid (BEH) Glycan column, 150 × 2.1 mm, 1.7 µm particles on a Waters Acquity UPLC H-Class instrument (both Waters, Milford, MA, USA). Separation was achieved in 30 min with a linear gradient of ACN and 50 mM ammonium formate (pH 4.4) at 560 µL/min. Label was detected at excitation/emission wavelengths λ_ex_ = 330 nm and λ_em_ = 420 nm, respectively. Glucose unit (GU) values were assigned after external calibration fitting a fifth-order polynomial to a 2AB-labeled dextran ladder [26].

### 2.4. Weak Anion Exchange Liquid Chromatography and Exoglycosidase Digestion of 2-Aminobenzamide Labeled Glycans (HILIC-FLD(2AB)/WAX)

WAX was performed using a 10 μm Biosuite DEAE (7.5 mm × 75 mm) column (Waters) on a 2795 Alliance separations module with a Waters 2475 fluorescence detector (see also previous publication [15]). Eluent A was 20% ACN, and eluent B was 25 mM ammonium acetate buffer adjusted to pH7.0. Samples were injected in water and subjected to a linear gradient of 100%A to 100%B over 50 min at a flow rate of 0.75 mL/min. Collections were made for neutral species N1 (2.0–4.9 min) and charged species S1–S4 (7.5–13.0 min, 13.0–18.8 min, 18.8–26.0 min and 26.0–45.0 min). A fetuin *N*-glycan standard was used for calibration.

All enzymes for the exoglycosidase digestion were purchased from ProZyme (San Leandro, CA, USA) or New England Biolabs (Ipswich, MA, USA). The 2AB-labeled glycans were digested in a volume of 10 μL for 18 h at 37 °C in 50 mM sodium acetate buffer, pH 5.5, using arrays of the following enzymes: *Arthrobacter ureafaciens* sialidase (ABS, EC 3.2.1.18, Prozyme), 0.5 U/mL; bovine testes β-galactosidase (BTG, EC 3.2.1.23, Prozyme), 1 U/mL; N-acetylneuraminate glycohydrolase (NAN1, EC 3.2.1.18, Prozyme), 5 U/mL; bovine kidney α-fucosidase (BKF, EC3.2.1.51, Prozyme), 1 U/mL; β-N-acetylglucosaminidase cloned from *S. pneumonia*, expressed in *Escherichia coli* (GUH, EC 3.2.1.30, NEB), 4000 U/mL. After incubation, enzymes were removed via filtration through 10 kDa protein-binding EZ filters (Merck KGaA). *N*-glycans were then analyzed by using HILIC-FLD(2AB).

### 2.5. Sample Preparation for Hydrophilic Interaction Liquid Chromatography-Fluorescence Detection after Procainamide Labeling (HILIC-FLD(PROC))

A Hamilton Microlab STARlet liquid-handling robot was used for all steps, namely for releasing, labeling and cleaning up *N*-glycan samples ready for analysis. Reagents used for *N*-glycan release were obtained from QABio, Palm Desert, CA, USA (PNGase F, E-PNG01). Samples were buffer exchanged into PBS using Vivaspin 6 centrifugal concentrators (Satorius, Goettingen, Germany), aliquoted in quintuplicate into a Framestar 96-Well skirted PCR plate (4ti-0690, Azenta Life Sciences, Burlington, MA, USA) and dried down ready for PNGase F release. Samples were resuspended in water (17.5 µL), followed by the addition of reaction buffer (5 µL) and denaturation solution (1.25 µL). The plate was heat sealed (Pierce heat seal, 4ti-0531, Azenta Life Sciences) and denatured at 100 °C for 10 min. Following the denaturation, the samples were allowed to cool to room temperature, and samples were incubated with Triton X-100 (1.25 µL) and PNGase F (at 1:1 dilution with water) at 37 °C overnight. Released *N*-glycans were dried using vacuum centrifugation and then converted to aldoses by incubating with 1% formic acid (Merck KGaA) solution at room temperature for 45 min. The remaining proteins and enzymes were removed using a LudgerClean Protein Binding Membrane Plate (LC-PBM-96, Ludger, Oxford, UK) via the addition of the samples to the plate and washing twice with 100 µL water. The washes collected along with the filtrate were transferred to a Framestar 96-Well non-skirted PCR plate (4ti-0750, Azenta Life Sciences) and dried down in a vacuum centrifuge.

Glycans were labeled with procainamide using a LudgerTag Procainamide Glycan Labeling Kit with 2-picoline borane (LT-KPROC-VP24, Ludger). The procainamide dye solution (DMSO, acetic acid, procainamide, 2-picoline borane and water) was prepared following the manufacturer’s protocol and added to the samples. Labeling was conducted at 65 °C for one hour, and the labeled samples were then cleaned-up.

The clean up of the samples and removal of excess dye was performed using a Ludger Clean plate (LC-PROC-96, Ludger) following the manufacturer’s protocol. Briefly, the samples were added to the plate in ACN, washed 3 times with ACN (3 × 200 µL) and eluted in water (3 × 100 µL). Purified labeled *N*-glycans were stored at 4 °C until they could be processed. The samples were measured as described in the next paragraph.

### 2.6. Analysis of Procainamide-Labeled Glycans, Optionally with Mass Spectrometry

Samples and standards were analyzed by HILIC-FLD(PROC) coupled to electrospray ionization tandem mass spectrometry (LC-ESI-MS/MS). Procainamide labeled glycan samples were injected in 75% aqueous ACN, with an injection volume 25 µL. The samples were analyzed by using HILIC on an Ultimate 3000 UHPLC using a BEH-Glycan 1.7 µm, 2.1 × 150 mm column (Waters) at 40 °C with a fluorescence detector (λ_ex_ = 310 nm, λ_em_ = 370 nm) controlled by Bruker HyStar 3.2 and Chromeleon data software version 7.2. Eluent A was 50 mM ammonium formate made from LudgerSep N Buffer stock solution, pH 4.4 (LS-N-BUFFX40, Ludger), and eluent B was ACN (acetonitrile 190 for UV/gradient quality; Romil #H049). Gradient conditions were: 0–53.5 min, 24–49.0% A (0.4 mL/min); 53.5–55.5 min, 49.0–0% A (0.4–0.2 mL/min); 55.5–57.5 min, 100% A (0.2 mL/min); 57.5–59.5 min, 100–24% A (0.2 mL/min); 59.5–66.5 min, 24% A (0.2 mL/min); and 66.5–70 min, 24% A (0.4 mL/min). Chromeleon data software version 7.2 with a cubic spline fit was used to allocate GU values to peaks. Procainamide-labeled glucose homopolymer was used as a system suitability standard and an external calibration standard for GU allocation for the system. Samples were then analyzed using a Bruker amaZon Speed ETD electrospray mass spectrometer, which was coupled directly after the UHPLC-FD without splitting. The instrument scanned samples in the maximum resolution mode, positive ion setting, MS scan + three MS/MS scans, nebulizer pressure 14.5 psi, nitrogen flow 10 L/min and capillary voltage 4500 V. MS/MS was performed on three ions in each scan sweep with a mixing time of 40 ms. Mass spectrometry data were analyzed using the Bruker Compass DataAnalysis 4.1 software. LC-ESI-MS/MS chromatogram analysis was performed using Bruker Compass DataAnalysis 4.4 and GlycoWorkbench software (2.1, ‘open source’, London, UK). Structures were identified by comparing LC, MS and MS/MS data.

## 3. Statistical Analysis

All data were corrected using total area normalization after determination of the final peak list to obtain relative glycan abundances. Statistical analysis was performed in GraphPad Prism for Windows (version 9.3.1; GraphPad Software, San Diego, CA, USA). Descriptive statistics of the distribution of coefficients of variation (CVs) of the different analytes measured by using an individual method in a specific preparation are provided (Table 1 and Appendix A). Differences between methods in these distributions were assessed with a Kruskal–Wallis test with multiple comparison (Appendix A). Multiple testing correction for this test was performed by using the Benjamini–Hochberg method using an FDR of 5%. Differences in relative glycan abundances (Appendix A) were assessed with an unpaired *t*-test using Bonferroni multiple testing correction.

## 4. Results and Discussion

Glycoanalytical methods may face very different demands depending on the application. This in itself underlines the need for multiple orthogonal methods. Nonetheless, some critical parameters can be identified by which different methods may be compared. 1. Low variability is a prerequisite for allowing the identification of differences between samples; 2. Resolution helps to distinguish molecular species with differential functional impact, thus allowing us to judge the impact of observed differences; 3. While different results are to be expected for different methods based on their specific strengths and weaknesses, all methods suitable for a specific type of analysis should produce the same major conclusions. These three parameters will guide the comparison of the presented methods, complemented by the qualitative discussion of other relevant parameters.

The *N*-glycosylation analysis of EPO biopharmaceuticals presents a very specific challenge, so results need to be viewed in this specific context. Still, throughout development and the product lifecycle, demands may strongly vary, especially with respect to the balance between variability and resolution.

Extensive information on the method performance can be extracted from the literature that introduces the respective methods [4,17,25]. Nonetheless, methods are most fairly compared if presented with the same challenges. In this case, two biopharmaceutical formulations and a reference standard of EPO were chosen.

### 4.1. Uniqueness and Repeatability of Profiles

Figure 1 provides an overview of the general quality of the mass spectra or chromatograms obtained with each technique, exemplified for Aranesp. The same overview for the two other samples is presented in Appendix A. Each sample was analyzed in quintuplicate using each method, wherein each replicate was separately subjected to the entire workflow (though as one batch). From these figures, a first impression of the variability can be gained. Variability in the retention time, *m*/*z* and intensity domain was relatively limited in comparison to the respective peak width and height. Since much of the characterization can and has been checked against the literature (see below), major peaks can be used for internal calibration to further reduce errors in the time and/or *m*/*z* domain.

The variability in the intensity domain may be directly translated into variability with respect to quantitation. However, in our glycomics experiments, this variability was reduced in two ways during data processing. Using the integral of chromatographic or mass spectrometric peaks corrected for the interplay between peak width and height. Total area normalization corrected for variation in the response factors between different measurements. The remaining variability, expressed as the coefficient of variation (CV), was greatly influenced by the method. However, it was still variable between analytes since it is influenced by (relative) abundance, specific interferences, etc. Therefore, Table 1 provides an overview of the distribution of variability per method for abundant analytes. The distribution of variability was generally consistent for all EPO preparations. The number of analytes quantified when using the different methods varied greatly (Appendix A). The higher the number of unique compositions identified, the greater the variability, especially if very low abundant analytes were quantified. Therefore, we focus on the variability of abundant analytes in Table 1, namely all analytes above 1% relative abundance. Overall, the median CV of the abundant analytes was low (<5%; expect Eprex by MALDI-MS) in each method. All methods are thus suitable for high-precision erythropoietin glycosylation analysis. The two HILIC-FLD methods were overall comparable in this respect, with the HILIC-FLD(PROC) results showing slightly lower variability for Aranesp (Appendix A). Equally, both HILIC-FLD methods slightly outperformed the MALDI-MS regarding variability.

Even at first glance, the profiles obtained per method for the three EPO preparations were different since all spectra had a unique set of peaks that were quantified. However, since this may be partially due to edge effects, we compared only peaks for which an individual method yielded a value in every EPO preparation. All methods were able to distinguish the three samples with very high confidence (Appendix A). Though this is not a common task for glycoanalytical methods, normally, the results would be compared to an expected range, for example in batch release, and it can be seen as an approximation of such comparisons. The HILIC-FLD methods rely on their high precision to show differences in their respective glycan peaks. The MALDI-MS method makes up for its slightly lower precision by assessing more analytes, especially low abundant analytes that are more likely to show larger variation between products. This can be seen in Appendix A, where a higher percentage of peaks differed between products in the HILIC-FLD methods, while a higher absolute number of peaks differed in the MALDI-MS method. Consequently, while subtle differences or changes in the major species will be picked up earlier with the HILIC-FLD methods due to their higher precision, the MALDI-MS method will more easily identify larger changes in minor species as it quantifies more of them. In the HILIC-FLD methods, strong changes in minor peaks may be missed or misinterpreted due to peak overlap with major species or will at least require the identification of additional peaks.

### 4.2. Structural Resolution

The MALDI-MS method identified the highest number of glycan compositions for Eprex (73 species; Appendix A). With the effort of the additional separation and exoglycosidase digestion, the HILIC-FLD(2AB)/WAX method could distinguish almost as many species (63 species). However, due to this effort, it could not be applied to every replicate. Without this, the HILIC-FLD methods reached significantly less molecular resolution (39 species for HILIC-FLD(PROC)-MS). It should be noted that the HILIC-FLD(PROC) method relies on hyphenated electrospray mass spectrometry for structural confirmation. Consequently, quantitative profiling of every sample and of low-abundant analytes is best performed with the MALDI-MS method. Looking into more detail, we can identify some reasons, but we can also find a number of features whose resolution is limited to the FLD-based methods. The MALDI-MS method distinguished eight species containing NeuGc for Eprex, which were not resolved by the other methods. Furthermore, the MALDI-MS method resolved, in total, 40 different species carrying various amounts of sialic acid *O*-acetylation, as opposed to 14 (HILIC-FLD(2AB)/WAX) and 25 (HILIC-FLD(PROC)-MS). Interestingly, this was dependent on glycan size and relative abundance. While for smaller and more abundant glycans, for example H5N4F1S1 (FA2G2S1) and H7N6F1S4 (FA4G4S4), HILIC-FLD(2AB)/WAX identified more *O*-acetylation variants, for larger and less abundant glycans, MALDI-MS offered more species. The reasons lie in the isomer separation potential of the HILIC as well as differences in the resolving power. MALDI-MS only delivers the number of *O*-acetylations, while HILIC-FLD additionally distinguishes isomers. However, this results in division of the signal and a hyper-linear increase in complexity with glycan size for the HILIC-FLD, which pushes many signals below the limit of quantitation and/or beyond its ability to resolve all existing isomers at the same time. While it is difficult to directly compare the mass spectrometric concept of resolution to the separation concept of theoretical plates, a glance at Figure 1 confirms the notion that the signal width is narrower compared to the total space over which signals are distributed in the MALDI-MS spectrum than in the HILIC-FLD chromatograms. In combination with the higher theoretical complexity of the HILIC-FLD chromatograms, this explains why more species were quantifiable with the MALDI-MS. It is noteworthy that, due to the superior performance for smaller glycans, the HILIC-FLD(2AB)/WAX method reached the highest number of non-*O*-acetylated elemental compositions (41 versus 33 for MALDI-MS and 14 for HILIC-FLD(PROC)-MS), which is typically the strength of MS-based methods. In addition, HILIC-FLD(2AB)/WAX identified eight isomers of non-*O*-acetylated species, mainly due to the ability to distinguish between an additional antenna and a LacNAc repeat.

### 4.3. Comparability of Conclusions

Qualitatively and quantitatively, the results were well comparable between methods. All major species were identified as such in all methods, and their ranking by relative abundance was shared between the methods. For example, for Eprex this was (by sum of *O*-acetylation variants; Appendix A): H7N6F1S4/FA4G4S4 > H7N6F1S3/FA4G4S3 > H8N7F1S4/FA4G4L1S4 > H6N5F1S3/FA3G3S3 ≈ H8N7F1S3/(FA3G3L2S3 + FA4G4L1S3) > H9N8F1S4/FA4G4L2S4 > H5N4F1S2/FA2G2S2 > H6N5F1S2/(FA2G2L1S2 + FA3G3S2). Smaller quantitative differences arose from the differences in structural assignment by exoglycosidase digestion versus mass spectrometry, as well as due to the number of resolved species. The HILIC-FLD(PROC)-MS method resolved the lowest number of species, which mainly resulted in a higher relative abundance of the main composition H7N6F1S4/FA4G4S4 and its *O*-acetylated variants compared to the MALDI-MS method with the largest number of species. The rest of the profile was highly comparable. Likely, co-eluting structures added their signal to the abundance of the major species. The HILIC-FLD(2AB)/WAX method also showed a higher abundance of H7N6F1S4/FA4G4S4 variants compared to the MALDI-MS method but also showed lower abundances of H8N7F1S3/FA4G4L1S3 + FA3G3L2S3, H8N7F1S4/FA4G4L1S4 and H7N6F1S3/FA4G4S3 compared to both other methods. Thus, this was likely due to the exoglycosidase-based assignment in the HILIC-FLD(2AB)/WAX compared to the MS-based assignments of the other two methods. Indeed, a deep, expert look at the data revealed that incomplete exoglycosidase digestion of FA4G4S4AcX and FA4G4L1S4AcX caused a lack of confidence in the assignment of these peaks. This can result in the observed differences as these species are easily distinguishable when using MS-based assignment due to their largely different mass.

Importantly, all methods were able to distinguish the three products with great confidence (Appendix A) and to produce precise glycosylation profiles (Table 1). Consequently, all methods are suited to perform the characterization of EPO products and will reach comparable conclusions, both regarding the absolute glycan profile and in the relative comparison of products.

### 4.4. Other Relevant Aspects

One clear difference between the MALDI-MS and the HILIC-FLD methods is that the former produces only one data dimension. Therefore, assignment and quantitation are conducted in one step, significantly contributing to a fast workflow. However, this limits the potential to identify analytes. While accurate mass and isotopologue patterns provide some confidence, this is generally only sufficient for previously characterized types of samples. In contrast, the combination of GU values from the HILIC separation, the sialic acid-centered separation of the WAX and the exoglycosidase digestion are considered adequate approaches for the identification of unknowns. The combination of GUs and accurate mass used in the HILIC-FLD(PROC)-MS method occupies a middle ground between the other methods and, thus, a gray area of highly confident assignment and defendable identification. Which approach is most efficient will thus depend on the previous knowledge of the sample. For well-characterized products, GUs from the initial HILIC-FLD step of the respective protocol are usually sufficient. Some known unknowns, for example arising from batch variations, could be covered by using the MALDI-MS method. Larger changes in profiles or unknown unknowns would require extensive HILIC-FLD(PROC)-MS or HILIC-FLD(2AB)/WAX methods. Alternatively, collision-induced fragmentation can supplement the MALDI-MS platform in this respect [17].

### 4.5. Hands-on Time and Protocol Length

The HILIC-FLD(2AB) utilizes a more complex glycan release protocol, including initial reduction, denaturation, alkylation and proteolysis into glycopeptides. While this is generally more robust, especially in complex samples, simpler approaches used by the other approaches are likely sufficient, and therefore more efficient, in recombinant proteins. Fast approaches may be considered as an alternative but generally come with a high consumption of PNGase F [14].

Beyond glycan release, MALDI-MS features the shortest sample preparation protocol that can be completed in under a day with under three hours of hands-on time. The HILIC-FLD protocols require several days with a significant portion of hands-on time. HILIC-FLD(2AB)/WAX additionally requires time and effort for the exoglycosidase incubations. Notoriously, exoglycosidase-based workflows require significant skill and rigorous controls to achieve robust annotation/identification results. In contrast, HILIC-FLD(PROC)-MS does not require additional sample preparation. However, additional hands-on time can be mitigated by the use of robotics, as demonstrated for both HILIC-FLD methods. Robotics automation is also available for the MALDI-MS method [27]. Sample preparation for the MALDI-MS, especially the spotting of samples, requires a more specialized set of skills, resulting in a significant hurdle for the implementation of this method.

Additionally, regarding measurement time, the MALDI-MS greatly outperforms the HILIC-FLD methods with circa 30 s versus up to 50 min per sample. This difference is further exaggerated when considering the fractionation and multiple HILIC-FLD measurements associated with the HILIC-FLD(2AB)/WAX. In contrast, investment and operational costs are lower per time unit—but not necessarily per sample—for HILIC-FLD compared to MALDI-MS.

The advent of automated processing tools in the last decade has not only strongly facilitated all employed methods by greatly reducing the (hands-on) time spent in data processing,[28] and it has also efficiently reduced the previous disadvantage of MS-based methods arising from the higher complexity of the data.

### 4.6. Robustness

The higher intrinsic complexity of the MS-based approaches comes with increased troubleshooting needs. Additionally, sufficient expertise is easier to obtain and maintain for the HILIC-FLD methods. Nonetheless, all three methods can be performed precisely, accurately and robustly with respect to both quantitation and assignment. An advantage of the MALDI-MS method is that it does not rely on a chromatographic column as a critical consumable. Batch-to-batch variation, retention time changes as a function of the number of injections of samples, which slowly leads to broadening of glycan peaks, or even discontinuation of a product can result in a significant effort to warrant continued robustness, especially if column selectivity is noticeably affected.

The performance of the assignment workflows is critical to the robustness of the obtained quantitative glycosylation profiles. As suggested earlier, exoglycosidase digestions in combination with LC techniques is probably the most powerful but also most fragile of the approaches used herein. While delivering the greatest resolution, specifically in terms of isomeric structures, almost consequently, it is most likely to experience performance issues. Precise annotation relies on complete and selective removal of targeted features, which requires high and constant activity and the selectivity of the employed glycosidases. While it can already be challenging to source such an enzyme for each desired feature, enzyme batch-to-batch variation and dependence on precise incubation conditions need to be controlled and monitored. Furthermore, the fact that it cannot be performed for all samples limits the utility of using isomers for distinguishing products or batches.

HILIC-FLD of released glycans is still the gold standard and thus can be found in almost every laboratory dealing with glycan analysis. However, academic and even industrial laboratories are increasingly embracing mass spectrometry, even at a quality control level. Though this trend is as yet mainly seen for peptide analysis, intact mass measurement of glycoproteins is also being performed, for example for batch release. We therefore expect mass spectrometry to also see increasing use in the routine analysis of released glycans.

### 4.7. Outlook

Though many of the results and discussion points of this study will remain relevant, there are many recent developments in glycosylation analysis that will surely impact how the glycosylation of EPO is assessed in the future. Most notably, the wider availability of mass spectrometry methods enables new approaches and enhances current ones. For example, a preferable addition to the HILIC-FLD(2AB)/WAX method is to perform HILIC(2AB)-MS of the WAX and exoglycosidase fractions [15]. Furthermore, analysis of the level of intact EPO [10] on the glycopeptide level,[11,12] especially combining multiple levels (including released glycans) [10], will see a rise in popularity. This could potentially be catalyzed if the possibility of studying site-specific glycosylation revealed functional differences between the sites.

Structural identification, especially of isomers, using exoglycosidases is increasingly refined by the characterization of a large number of exoglycosidases with novel activities and selectivities, enabled by the use of metagenomics screening and the rising interest in the microbiome [29]. The MALDI-MS method can distinguish between sialic acid linkage isomers. Though not relevant for CHO cell-produced EPO, exclusively featuring α2,3-linked sialylation, analysis of EPO from alternative production systems may benefit from this feature.

Additionally, the separation of released glycans has seen encouraging developments that combine orthogonal—compared to each other and to HILIC—selectivities and excellent compatibility with MS [30,31]. More specifically, for combination with MALDI-MS, gas phase separations may offer an interesting way to allow the distinction of isomers [32].

On the side of data interpretation, the integrative use of the fluorescence and the mass spectrometric signal promises increased performance next to the integration of analysis levels [31].

## 5. Conclusions

All three profiling methods were able to distinguish the three EPO preparations with very high confidence. They showed low variability (median CVs < 5%), detected the same major species and very similarly ranked their relative abundances. Consequently, all studied methods are suitable for high-precision EPO *N-*glycosylation profiling.

HILIC-FLD methods are still the most accessible, precise and robust way to profile released glycans of EPO. For routine profiling, they can distinguish a sufficient number of abundant structures. However, when it comes to in-depth analysis, especially identification, the complexity of EPO glycosylation exceeds the possibilities of HILIC-FLD, and additional or alternative approaches are needed. The complex but extremely thorough approach of HILIC-FLD(2AB)/WAX with exoglycosidase digestion distinguishes the most types of isomers. However, the complexity limits sensitivity and greatly limits throughput, making it the method most suited for the initial identification of abundant structures. At the other extreme, MALDI-MS offers the fastest and most sensitive approach as it combines profiling and identification in one experiment but offers no isomer resolution. The speed makes it a very attractive choice for large sample sets. Due to the high sensitivity, more analytes can be quantified if one accepts the structural ambiguity of the compositions. HILIC-FLD(PROC)-MS shares the lower sensitivity with HILIC-FLD(2AB)/WAX, with an isomer resolution somewhere in between the other two methods. As it can be applied to every sample in a dataset, as opposed to HILIC-FLD(2AB)/WAX, which is simply too complex, the HILIC-FLD(PROC)-MS method excels at quantitatively addressing the structural complexity of abundant analytes in medium-sized sample sets. A challenge prominent in EPO, and often neglected in other sample types,[33] is sialic acid *O*-acetylation, which greatly enlarges complexity, especially in tri- and tetraantennary glycans due to the combinatorial explosion of numeric and isomeric *O*-acetylation variants. All methods were able to deal with this challenge but showed interesting differences in coverage in line with their capabilities. With increasing awareness of the relevance of this challenging modification, we expect to see an even better coverage of these variants in all methods.

New developments can address some of the less favorable aspects of each method, but it remains to be seen how much of the advantages have to be traded in.

## Figures and Tables

**Figure 1 biomolecules-14-00125-f001:**
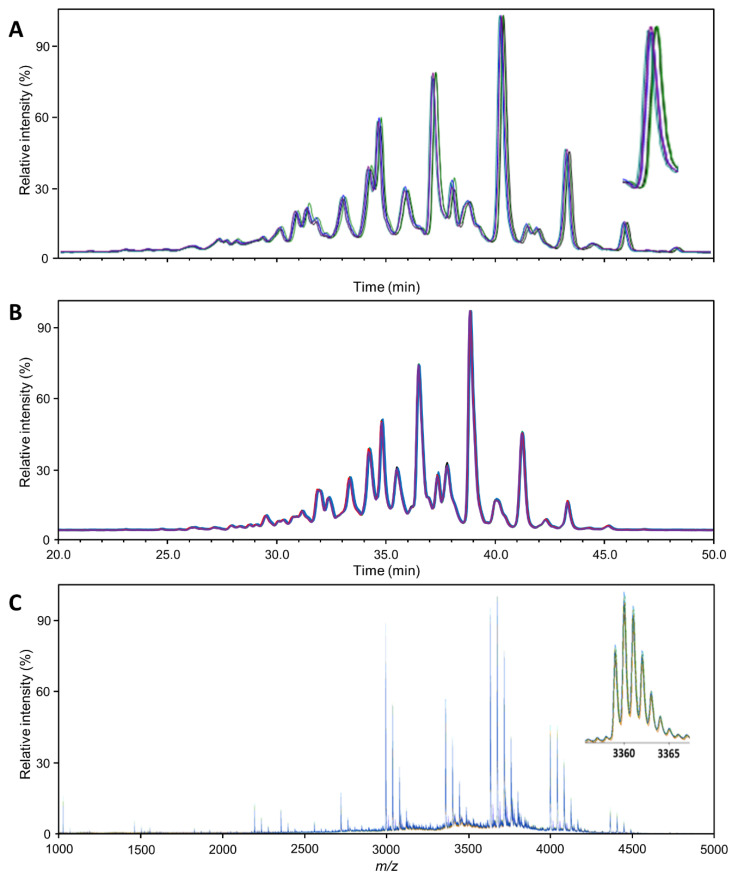
Profile comparability: Overlay of the 5 Aranesp traces (replicates; different colors), normalized to the highest peak. (**A**) HILIC-FLD(2AB); (**B**) HILIC-FLD(PROC); (**C**) MALDI-MS. The magnifications show the general appearance and variability of a single analyte signal. The regular patterns following each major analyte (e.g., H7N6F1S4) are caused by sialic acid *O*-acetylation variants. The insert in (**C**) shows the isotopologue pattern of a single analyte ion ([M + Na]^+^).

**Table 1 biomolecules-14-00125-t001:** Precision of glycosylation profiling. Descriptive statistics for the distribution of CVs of all analytes above 1% relative abundance.

	Aranesp			PharmEPO			Eprex		
	HA	HP	MS	HA	HP	MS	HA	HP	MS
number of analytes	16	23	20	16	21	19	17	25	22
Median CV	2.3	1.7	2.9	2.5	1.4	4.3	1.7	2.2	8.7
(95% CI)	1.9–3.4	0.9–2.1	2.4–3.6	1.1–3.0	0.9–2.0	3.4–6.2	1.1–2.2	1.4–2.5	5.5–13.9
IQR	1.9–3.4	0.9–2.2	2.4–3.8	1.2–2.9	0.9–2.1	3.4–6.2	1.0–2.5	1.4–2.7	5.4–14.0
Min-Max	1.0–5.0	0.5–3.7	1.3–7.5	0.6–4.4	0.4–4.0	1.9–8.5	0.5–4.8	0.3–5.3	5.4–24.3

HA = HILIC-FLD(2AB); HP = HILIC-FLD(PROC); MS = MALDI-MS. IQR = interquartile range.

## Data Availability

The data presented in this study are available on request from the corresponding author. The data are not publicly available due to expected limited public interest in the raw data.

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
