# Peer review of "Erythropoietin N-glycosylation of Therapeutic Formulations Quantified and Characterized: An Interlab Comparability Study of High-Throughput Methods"

_biomolecules, 2024, doi:10.3390/biom14010125_

Round 1

Reviewer 1 Report

Comments and Suggestions for Authors

The authors report comparison of the EPO N-glycosylation analysis (qualitative and quantitative) by using three of the most used methods: HILIC-FLD of the released N-glycans after 2AB derivatization, HILIC-FLD of the released N-glycans after procainamide labeling and MALDI-MS of the released N-glycans after sialic acids lactonization.

Authors are well known scientist in the field, experimental procedures are well done and described (except in one case) and the manuscript deserves publication on biomolecules after minor revisions.

1)    MALDI-MS preparation procedure is not well described. Even if it is in the citation 15 (Sci rep 7, 5324 (2017)) a more complete description of the method in this manuscript is preferable, also for conformity with the other two techniques with which it is compared

2)    In the Abstract line 27 “only HILIC-FLD covered isomerism”. I think it is better to specify which isomerism. In fact, in the case of 2-3 2-6 sialic acids, Maldi-MS is able to distinguish these two isomers

Author Response

Comment 1)    MALDI-MS preparation procedure is not well described. Even if it is in the citation 15 (Sci rep 7, 5324 (2017)) a more complete description of the method in this manuscript is preferable, also for conformity with the other two techniques with which it is compared

Answer 1): We thank the reviewer for alerting us to this mismatch in the extend of the descriptions between the methods. We have extended the description of the MALDI-MS procedures accordingly.

Comment 2)    In the Abstract line 27 “only HILIC-FLD covered isomerism”. I think it is better to specify which isomerism. In fact, in the case of 2-3 2-6 sialic acids, Maldi-MS is able to distinguish these two isomers

Answer 2: In the abstract, we now specify the isomerism encountered. Though powerful, the distinction of sialic acid linkage isomers is not relevant to this study as CHO cells – the dominant biotechnological platform for EPO production and source of all studied products – produce alpha 2-3 linked sialylation exclusively. We have added a sentence to the “Outlook” regarding the relevance sialic linkage isomerism distinction.

Reviewer 2 Report

Comments and Suggestions for Authors

The submitted manuscript covers three common methods to assay N-glycan focused critical quality attributes of the biotherapeutic erythropoietin. 

General questions 

- I can't remember, does EPO have O-glycosylation? if it does I would briefly mention it when glycopeptide and intact analysis is mentioned in the introduction. I agree it's not a focus of this paper but it should be mentioned for readers not familiar with protein glycosylation.

-Other glycosylation analysis methods include permethylation and Mass Spectrometry, as well as a multitude of kit based labeling kits from vendors like Agilent or Waters. These also include multiple "quick labeling" chemistry that is different than reductive amination and is frequently utilized for this type of work. 

- Will the raw data be provided?

- Can the authors speak to whether the cleanup methods SPE kits would impact the glycosylation observed? Eg impacts on sialylation or LacdiNac

Specific comments:

-Page 1 line 31: "less experimental expanse"? Do you mean expense or the expansiveness of the workflow? 

-Figure 1 C: In the MALDI data what are the enveloped m/zs? are they adducts? 

-

Author Response

General questions 

Comment1: - I can't remember, does EPO have O-glycosylation? if it does I would briefly mention it when glycopeptide and intact analysis is mentioned in the introduction. I agree it's not a focus of this paper but it should be mentioned for readers not familiar with protein glycosylation.

Answer1: We have made the corresponding additions (page 1 lines 39-40 and page 2 lines 55-56) and referred to the Hua et al review again which provides an excellent overview also in this respect.

Comment2: -Other glycosylation analysis methods include permethylation and Mass Spectrometry, as well as a multitude of kit based labeling kits from vendors like Agilent or Waters. These also include multiple "quick labeling" chemistry that is different than reductive amination and is frequently utilized for this type of work. 

Answer2: We have made a few additions to the third paragraph of the introduction to provide a more balanced birds-eye view, gladly using the reviewers suggestions. We hope that the reviewer agrees that a more in-depth discussion is not desirable due the excellent reviews on this topic as exemplified in the first line of the paragraph.

Comment3: - Will the raw data be provided?

Answer3: We do not expect much interest in the raw data, as, in our view, the potential for reanalysis is limited (for example, exclusive focus on released N-glycans and already provided extensive structural analysis). Therefore, deposition of raw data was not intended.

Comment4: - Can the authors speak to whether the cleanup methods SPE kits would impact the glycosylation observed? Eg impacts on sialylation or LacdiNac

Answer4: Our data does not suggest an impact of the SPE steps on the glycosylation profiles observed. On the contrary, the fact that despite diverse SPE stationary and mobile phases, all methods obtained the same overall glycosylation profiles confirms that the SPE steps do not introduce major biases. Small differences (see for example Table S4) are more likely due to differences in resolution, as discussed under “4.3. Comparability of conclusions”.

Specific comments:

Comment5: Q-Page 1 line 31: "less experimental expanse"? Do you mean expense or the expansiveness of the workflow? 

Answer5: The statement concerns the amount of work involved in this workflow rather than the cost as we did not attempt a cost analysis. We hope the new phrasing is clearer.

Comment6: -Figure 1 C: In the MALDI data what are the enveloped m/zs? are they adducts? 

Answer6: The ‘clusters’ of peaks with a regular pattern in Figure 1C, that bear resemblance to a situation where an analyte is split into multiple adduct signals, derives from the sialic acid O-acetylation variants. Since the three sourced products, especially Aranesp are highly O-acetylated, this pattern is comparably extensive. The insert shows the full isotopologue pattern of a single analyte ion ([M+Na]+; no adducts). Both aspects have been clarified in the figure caption.

Reviewer 3 Report

Comments and Suggestions for Authors

In this research article, O’Flaherty et al. investigated the characterization and quantification of erythropoietin N-glycosylation of therapeutic formulations with an interlab comparability  study of high-throughput methods. Erythropoietin is an excellent model glycoprotein to study diverse N-glycan structures. Especially its rich N-glycan structures belonging to three main types of glycan; high mannose, complex and hybrid hinder accurate mass spectrometry analysis. In this research, the authors tested the following methods for this purpose;

-Hydrophilic interaction liquid chromatography with fluorescence detection after 2-aminobenzamide labeling (HILIC-FLD(2AB)) and optional weak anion ex-change chromatography (WAX) fractionation and exoglycosidase digest,

- HILIC-FLD after procainamide labeling (PROC) optionally coupled to electrospray ionization-MS and

-Matrix-assisted laser desorption ionization time-of-flight mass spectrometry (MALDI-MS).

They found that all techniques showed  good precision and were able to differentiate the unique N-glycosylation profiles of the various EPO  preparations and HILIC-FLD showed higher precision while MALDI-TOF-MS covered the most analytes. However, only HILIC-FLD covered isomerism. Although these results can be predicted based on the literature, the findings are extremely important field since N-glycan analysis method can be chosen based on the purpose. For example, as the authors mentioned. For routine profiling, HILIC-FLD methods are  more accessible and cover isomerism in major structures, while MALDI-MS covers more minor analytes with an attractively high throughput. For in-depth characterization, MALDI-MS and HILIC-FLD (2AB)/WAX give a similar amount of notably orthogonal information. HILIC-FLD(PROC)-MS is attractive for covering isomerism of major structures with significantly less experimental expanse compared to HILIC-FLD(2AB)/WAX.

In general, the study is well planned and organize to understand the N-glycan characterization differences with the major techniques. The results are statistically analyzed properly to show the differences.

Author Response

We thank the reviewer for taking the time to access our study. As we have not identified any specific comments, we addressed the indicated potential for improvements in connecting results and conclusions in the context of the suggestions of the other reviewers.